# Intermolecular diastereoselective annulation of azaarenes into fused N-heterocycles by Ru(II) reductive catalysis

He Zhao[1,4], Yang Wu[1,4], Chenggang Ci[2], Zhenda Tan[1], Jian Yang[1], Huanfeng Jiang [1], Pierre H. Dixneuf[3] & Min Zhang [1✉]

Derivatization of azaarenes can create molecules of biological importance, but reductive functionalization of weakly reactive azaarenes remains a challenge. Here the authors show a dearomative, diastereoselective annulation of azaarenes, via ruthenium(II) reductive catalysis, proceeding with excellent selectivity, mild conditions, and broad substrate and functional group compatibility. Mechanistic studies reveal that the products are formed via hydride transfer-initiated $\beta$-aminomethylation and $\alpha$-arylation of the pyridyl core in the azaarenes, and that paraformaldehyde serves as both the C1-building block and reductant precursor, and the use of Mg(OMe)$_2$ base plays a critical role in determining the reaction chemo-selectivity by lowering the hydrogen transfer rate. The present work opens a door to further develop valuable reductive functionalization of unsaturated systems by taking profit of formaldehyde-endowed two functions.

[1] Key Lab of Functional Molecular Engineering of Guangdong Province, School of Chemistry and Chemical Engineering, South China University of Technology, Guangzhou 510641, China. [2] Key Laboratory of Computational Catalytic Chemistry of Guizhou Province, Department of Chemistry and Chemical Engineering, Qiannan Normal University for Nationalities, Duyun 558000, China. [3] University of Rennes, ISCR, UMR CNRS 6226, 35000 Rennes, France. [4] These authors contributed equally: He Zhao, Yang Wu. ✉email: minzhang@scut.edu.cn

Azaarenes constitute a class of ubiquitously distributed substances applied in numerous fields of science and technology[1,2]. The development of new strategies enabling efficient and selective transformation of weakly reactive azaarenes into functional frameworks is of important significance, as they not only pave the avenues to access novel functional products, but also enrich the synthetic connotation of the azaarenes. To date, except the well-established electrophilic substitution utilizing azaarenes as the nucleophiles under harsh conditions[3,4], the recently emerged C–H bond activation/functionalization has offered many desirable ways for structural modification of the azaarenes[5–7]. In comparison with these aromaticity-retaining transformations, only a handful examples focused on dearomative coupling of active indole derivatives[8–11], whereas dearomative functionalization of inert pyridine-fused azaarenes (e.g., quinolines, isoquinolines, naphthyridines, phenanthroline, etc.)[12–14] has been scarcely explored.

In recent years, hydrogen transfer-mediated coupling reactions have emerged as appealing tools in the production of various functional products, since there is no need for high pressurized $H_2$ and elaborate experimental setups. For instance, in addition to the well-known reductive amination applied for amine syntheses[15,16], several groups such as Beller[17,18], Kempe[19,20], Kirchner[21,22], Liu[23–25], and others[26–29] have applied borrowing-hydrogen strategy to alkylate amines and the $\alpha$-site of carbonyl compounds with alcohols. Krische has demonstrated elegant contributions on the linkage of alcohols/carbonyls with unsaturated C–C bonds[30–32]. Bruneau et al. have achieved $\beta$-C(sp$^3$)–H alkylation of N-alkyl cyclic amines[33,34]. The Li group has converted phenols into synthetically useful amines[35,36]. Our group has reported a reductive quinolyl $\beta$-C–H alkylation with a low-active heterogeneous cobalt catalyst[37]. Later, Donohoe et al. have demonstrated interesting examples on the $\beta$-functionalization of azaarenes[38–40]. Despite these important advances, the strategy incorporating a tandem coupling sequence into the reduction of azaarenes remains to date a challenge due to the difficulty in controlling the reaction selectivity: on one hand, the azaarenes tend to undergo direct hydrogenation to form non-coupled cyclic amines under catalytic reduction conditions, on the other hand, it is hard to selectively transfer hydrogen only to one specific sites among different substrates.

Here, we conceived that, through an initial N-alkylation of azaarenes $\mathbf{A'}$ with bromoalkanes to form azaarenium salts[41,42], a solution to achieve the desired synthetic purpose would be offered: (i) the combination of a suitable metal catalyst (M) and hydrogen donor (HD) forms reductive metal hydride species [HM$^n$X] in-situ, which allows hydride transfer (TH) to the azaarenium salts $\mathbf{A}$ to generate allylic amine $\mathbf{int\text{-}1}$ and its tautomer N-alkyl enamine $\mathbf{int\text{-}2}$ (Fig. 1a). Such an enamine ($\mathbf{int\text{-}2}$) has higher $\beta$-reactivity in trapping electrophiles than its −NH counterpart and lowers the formation of non-coupled cyclic amine $\mathbf{A''}$. (ii) It is relatively difficult to reduce electron-rich enamine $\mathbf{int\text{-}2}$ to the undesired cyclic amine $\mathbf{A''}$.

Based on the above idea, we here report a dearomative annulation reaction of azaarenium salts $\mathbf{A}$ with aniline derivatives $\mathbf{B}$ and paraformaldehyde (Fig. 1b) under ruthenium(II) reductive catalysis, which offers a general way for diastereoselective construction of fused $syn$-N-heterocycles $\mathbf{P}$ featuring promising structural motifs of teterahydroquinoline and hexahydro-1,6-naphthyridine that are frequently found in natural alkaloids[43,44] and biomedical molecules[45–47], as exemplified by the leading anesthesia drug taripiprazole $\mathbf{1}$, active composition $\mathbf{2}$ used for treating EP1 receptor-mediated diseases[45], PXR agonist $\mathbf{3}$[46], and anticancer agents $\mathbf{4}$ (Fig. 1c)[47].

## Results

**Investigation of reaction conditions**. We commenced our studies by performing the reaction of N-benzyl quinolinium bromide $\mathbf{A1}$, N-ethylaniline $\mathbf{B1}$, paraformaldehyde, and base in MeOH at 65 °C for 18 h by employing [RuCl$_2$ ($p$-cymene)]$_2$ as the catalyst. Among various bases and acids tested, Mg(OMe)$_2$ exhibited the best chemo-selectivity since there is no formation of by-product N-benzyl tetrahydroquinoline $\mathbf{A1''}$ (Table 1, entries 1, 4 and Supplementary Table 1 in the Supplementary Information (SI)). The absence of catalyst or base failed to yield product $\mathbf{P1}$ (entries 5, 6), showing that both of them are indispensable for the product formation. Then, we screened several other metal catalysts applied frequently in hydrogen transfer reactions (see Supplementary Table 1 in SI). The results showed that Ir(I) or Ir(III) catalysts were also applicable, but the base metal catalysts (Co, Fe, Mn, and Ni) were totally ineffective for the transformation (entries 7, 8 and Supplementary Table 1). Here, we chose the cost-effective [Ru($p$-cymene)Cl$_2$]$_2$ as the preferred catalyst to further evaluate the solvents and temperatures, it showed that methanol and 55 °C were more preferable (entries 9, 10). Decrease of the base or (CH$_2$O)$_n$ amount diminished the product yields (entries 11, 12). Thus, the optimal yield of product $\mathbf{P1}$ was obtained when the reaction in methanol was performed at 55 °C for 18 h by using the combination of [Ru($p$-cymene)Cl$_2$]$_2$ and Mg(OMe)$_2$ (entry 10). Interestingly, the use of Mg(OMe)$_2$ base always resulted in excellent selectivity in affording product $\mathbf{P1}$ (entries 3, and 7, 12).

**Substrate scope**. With the availability of the optimal reaction conditions (Table 1, entry 10), we then assessed the substrate scope of the newly developed synthetic protocol. As shown in Fig. 2, various quinolinium salts $\mathbf{A}$ ($\mathbf{A1–A21}$, see Supplementary Fig. 1 in SI for their structures) in combination with N-ethylaniline $\mathbf{B1}$ and paraformaldehyde were evaluated. Gratifyingly, all the reactions underwent smooth reductive annulation and furnished the desired fused N-heterocycles in reasonable to excellent isolated yields with excellent $syn$-diastereoselectivity ($\mathbf{P1–P21}$, d.r. > 20:1). The structure of compound $\mathbf{P1}$ was confirmed by X-ray crystallography diffraction and NOESY spectrum (Supplementary Fig. 3–5 and Supplementary Table 2 in SI). The application of 1,5-dibromopentane for di N-alkylation generated the intramolecular alkyl-linked product $\mathbf{P21}$ in a good yield. Noteworthy, a variety of functionalities (e.g., −Me, −OMe, −SPh, amido, −F, −Cl, −Br, ester, −CF$_3$, −NO$_2$, alkenyl, and alkyl) on the quinolinium salts were well tolerated, and their electronic properties affected the product formation to some extent. Interestingly, no reduction of the nitro and alkenyl groups was observed ($\mathbf{P17}$ and $\mathbf{P18}$), and the halo-substrates also did not undergo hydrodehalogenation, indicating that the reaction proceeds in a chemoselective manner. In general, quinolines bearing an electron-donating group ($\mathbf{P2–P7}$, and $\mathbf{P13–P14}$) afforded relatively higher product yields than those having an electron-withdrawing group ($\mathbf{P8–P10}$, and $\mathbf{P12}$), presumably because the electron-rich quinolinium salts can result in more reactive enamine intermediates that are beneficial to the electrophilic coupling process (Fig. 1a). The retention of these functionalities offers the potential for post-functionalization of the obtained products.

Next, we turned our attention to the synthesis of structurally diversified products by variation of both azaarenes $\mathbf{A'}$ and anilines $\mathbf{B}$. First, a series of N-alkyl anilines ($\mathbf{B2–B18}$, see Supplementary Fig. 2 in SI for their structures) in combination with quinolinium salt $\mathbf{A1}$ were tested. As illustrated in Fig. 3, all the reactions efficiently afforded the desired product in moderate to excellent

**Fig. 1 Diastereoselective construction of functional polycyclic N-heterocycles by hydride transfer-initiated intermolecular annulation of the azaarenes. a** The formation of N-alkyl enamine **int-2**. **b** ruthenium-catalyzed dearomative annulation reaction of azaarenium salts A with aniline derivatives B and paraformaldehyde. **c** Selected drugs and bioactive molecules.

### Table 1 The optimization of reaction conditions[a].

| Entry | Catalyst | Base | P1 (%)[b] | A1″ (%)[b] |
|---|---|---|---|---|
| 1 | [Ru(p-cymene)Cl₂]₂ | K₂CO₃ | <10 | 8 |
| 2 | [Ru(p-cymene)Cl₂]₂ | t-BuOK | < 10 | 42 |
| 3 | [Ru(p-cymene)Cl₂]₂ | Mg(OMe)₂ | 90 | 0 |
| 4 | [Ru(p-cymene)Cl₂]₂ | MeOK | < 10 | 53 |
| 5 | - | Mg(OMe)₂ | 0 | 0 |
| 6 | [Ru(p-cymene)Cl₂]₂ | – | 0 | 0 |
| 7 | [Cp*IrCl₂]₂ | Mg(OMe)₂ | 85 | trace |
| 8 | [IrCl(COD)]₂ | Mg(OMe)₂ | 74 | trace |
| 9 | [Ru(p-cymene)Cl₂]₂ | Mg(OMe)₂ | (0, 0, 61)[c] | (0, 0, 0)[c] |
| 10 | [Ru(p-cymene)Cl₂]₂ | Mg(OMe)₂ | (58, 91, 86)[d] | (0, 0, 0)[d] |
| 11 | [Ru(p-cymene)Cl₂]₂ | Mg(OMe)₂ | (33, 25)[e] | (0, 0)[e] |
| 12 | [Ru(p-cymene)Cl₂]₂ | Mg(OMe)₂ | (42, 30)[f] | (0, 0)[f] |

Cp*: 1,2,3,4,5-pentamethylcyclopentadiene, cod: 1,5-cyclooctadiene, DMF: N,N-dimethylformamide.
[a]Unless otherwise stated, the reaction in MeOH (1 mL) was performed with **A1** (0.2 mmol), **B1** (0.2 mmol), cat. (1 mol%), base (0.75 eq), (CH₂O)ₙ (10 eq) at 65 °C for 18 h under N₂ protection.
[b]NMR yield by using anisole as the internal standard.
[c]Yields are with respect to the use of DMF, 1,4-dioxane, ethanol as the solvent, respectively.
[d]Yields are with respect to the temperature at 45 °C, 55 °C, 75 °C, respectively.
[e]Yields are with respect to 0.5 and 0.3 eq of Mg(OMe)₂, respectively.
[f]Yields are with respect to 8 and 5 eq (CH₂O)ₙ, respectively.

isolated yields with exclusive *syn*-selectivity (**P22–P36**, d.r. > 20:1). The electronic properties of the substituents on the benzene ring of the anilines significantly affected the product formation. Especially, anilines containing electron-donating groups (**P22–P23**, **P27** and **P35**) gave much higher yields than those with electron-withdrawing groups (**P24–P25**). This observation is attributed to electron-rich anilines favoring the electrophilic coupling process during the formation of the products. In addition to N-alkyl anilines, diarylamine **B13** also served as an effective coupling partner, affording the N-aryl product **P33** in a moderate yield. As expected, primary anilines were not applicable for the transformation, as they

easily reacted with formaldehyde to form aminals. Interestingly, tetrahydroquinolines (**B14** and **B15**) and 2,3,4,5-tetrahydro-1H-benzo[b]azepine (**B16**), two specific aniline derivatives, also underwent efficient multicomponent annulation to afford the polycyclic products (**P34–P35**, **P38** and **P41**). In addition to quinolines, other azaarenes, such as 1,8-naphthyridines (**A22–A25**), thieno[3,2-b] pyridine **A26**, 1,7-phenanthroline **A27**, 1,10-phenanthroline **A28**, and 5-substituted isoquinolines (**A29** and **A30**) were also amenable to the transformation, delivering the desired products in an efficient manner (**P37–P49**, d.r. > 20:1), these examples demonstrate the capability of the developed chemistry in the functionalization of

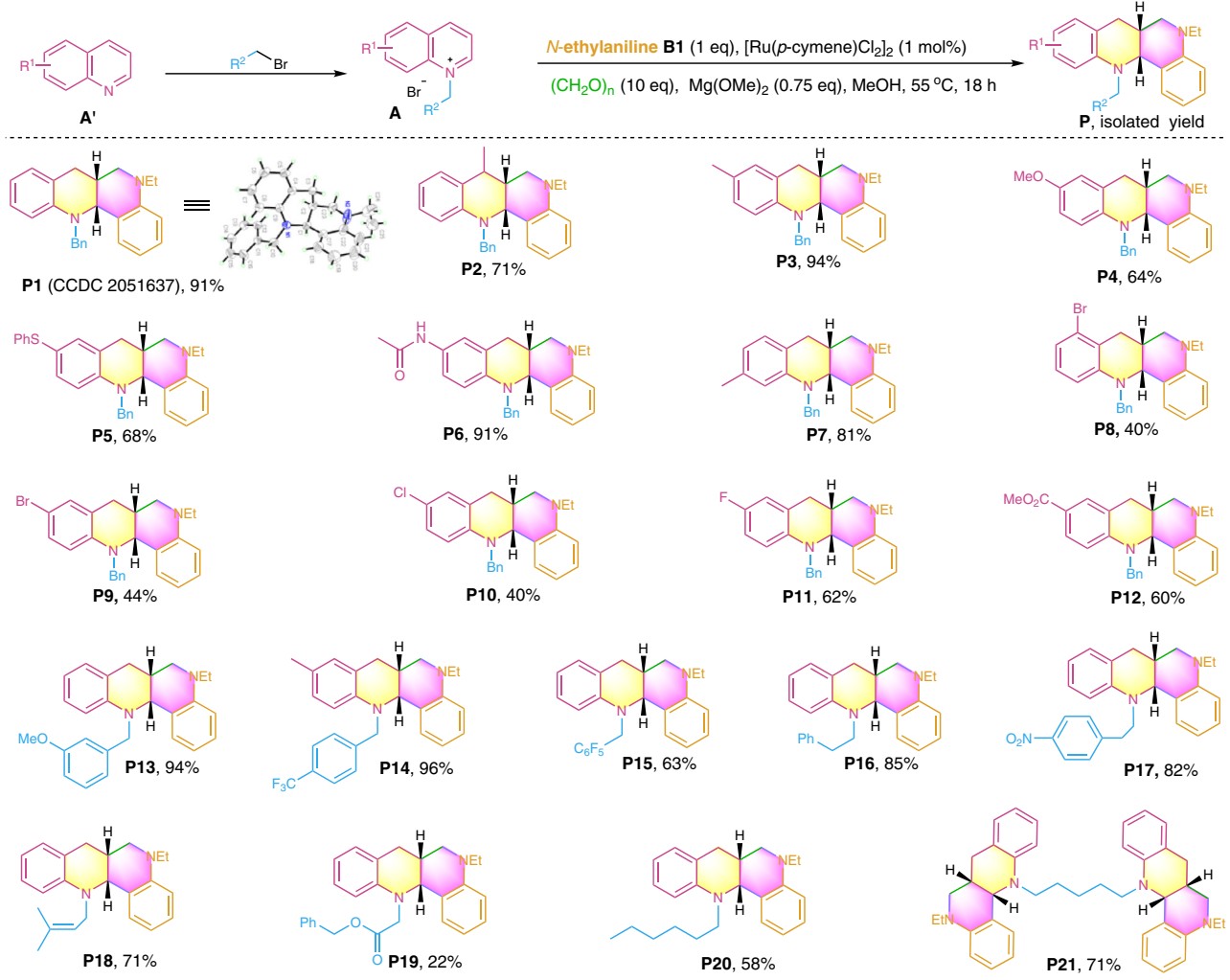

**Fig. 2 Diastereoselective construction of fused N-heterocycles P1 − P21 by variation of quinolines.** Reactions were conducted on a 0.2 mmol scale under the standard conditions. Isolated yields are reported.

pyridine-containing azaarenes including the N-bidentate ligands (**P37–P42**, **P47**). Unfortunately, more challenging pyridine derivative failed to yield the desired product, only β-aminoalkyl product (**P50**) was obtained.

Noteworthy, 5-substituted isoquinolines afforded the desired annulation products (Fig. 3, **P48** and **P49**), whereas 5-nonsubstituted isoquinolines generated products **P** by installing an additional methyl group at the β-site of the N-heteroaryl reactants, and all the products are produced with exclusive *syn*-diastereoselectivity (d.r. > 20:1, Supplementary Fig. 6). As shown in Fig. 4, N-benzyl isoquinolinium salts were firstly employed to couple with paraformaldehyde and N-ethylaniline **B1**. All the reactions gave rise to the desired annulation products in moderate to excellent yields upon isolation (**P51–P63**). Then, the transformation of secondary anilines including the N-alkyl and N-aryl ones was evaluated. Gratifyingly, all these anilines smoothly coupled with N-benzyl quinolium salt **A1** and paraformaldehyde, delivering the annulation products in reasonable to high yields (**P64–P76**). Similar to the results described in Figs. 2 and 3, various functionalities on both isoquinolium salts and anilines are well tolerated (−Bn, −Et, −Me, −F, −Cl, −Br, boronic ester, −SO2Me, −n-hexyl, −OMe, −CF3, −CO2Me, alkenyl, cyclohexyl, and i-propyl). The substituents on the aryl ring of the isoquinoline salts have little influence on the product formation, whereas the substituents of the anilines significantly

affected the product yields. Generally, aniline bearing an electron-donating group afforded higher yields (e.g., **P64–P66** and **P70–P73**) than those of anilines with an electron-withdrawing group (e.g., **P67–P69** and **P74**), suggesting that the reaction involves an electrophilic coupling process. Benzocyclic amines (1,2,3,4-tetrahydroquinoxaline, 1,2,3,4-tetrahydroquinoline, and 2,3,4,5-tetrahydro-1H-benzo[b]azepine) and N1-isopropyl-N4-phenylbenzene-1,4-diamine also served as effective coupling partners, affording the polycyclic products in moderate to high yields (**P77–P80**). These examples show the practicality of the developed chemistry in the construction of structurally complex polycyclic N-heterocycles.

**Synthetic applications**. Further, we explored the synthetic applications of the developed chemistry. As shown in Fig. 5a, 6-ester quinolinium salts, arising from initial esterification of 6-carboxylic quinoline and subsequent pretreatment with benzyl bromide, efficiently reacted with aniline **B1** and paraformaldehyde to afford products **P81** and **P82** (d.r. > 20:1), which are the analogs of analgesic[48] and the agents used for antioxidation and antiproliferation[49], respectively. Through successive amidation and formation of N-benzyl heteroarenium salt, 6-amino quinoline was efficiently transformed in combination with aniline **B1** into camphanic amide **P83** (d.r. > 20:1, Fig. 5b), an agent capable

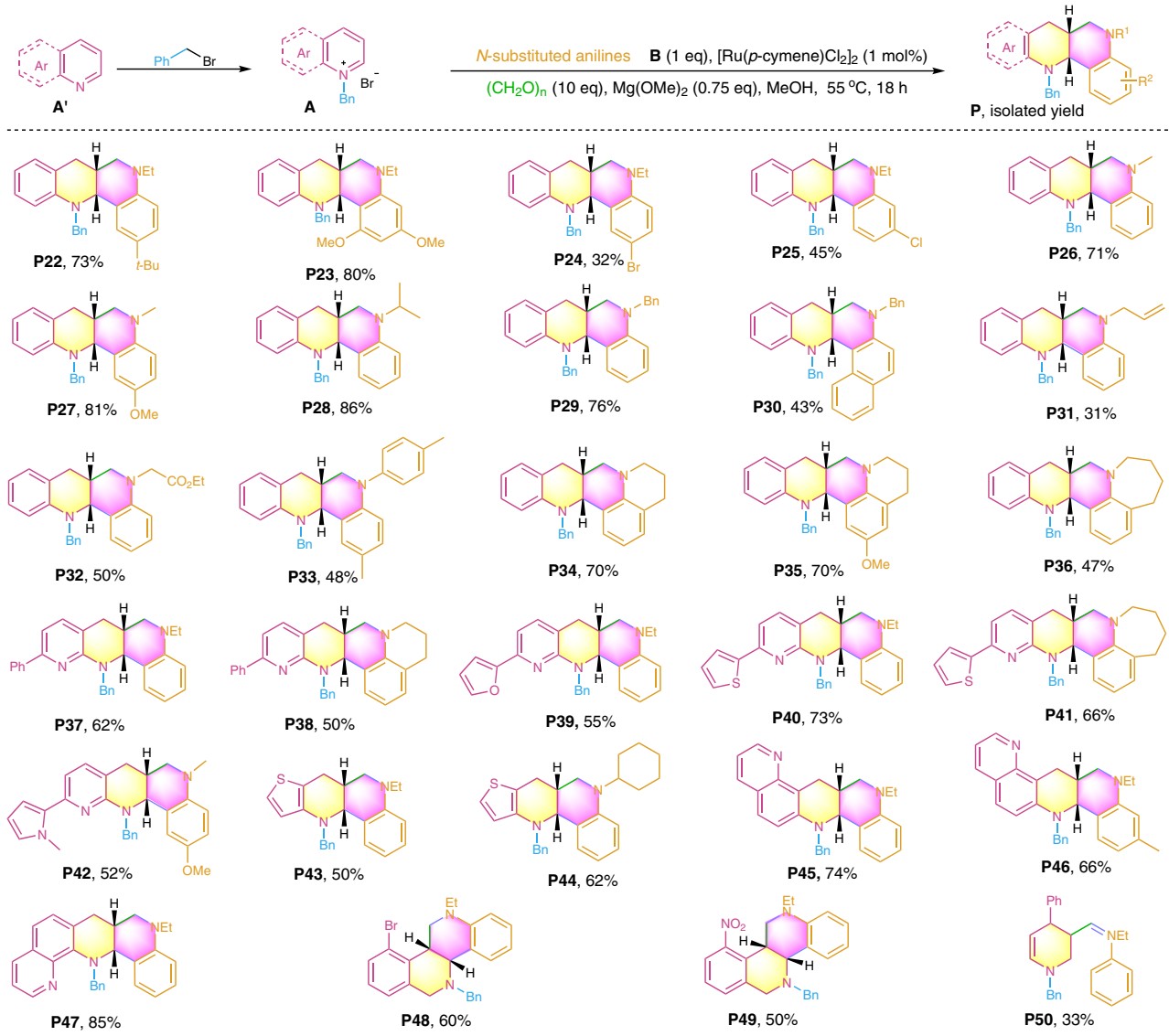

**Fig. 3 Diastereoselective access to fused N-heterocycles P22–P50 by variation of both azaarenes and anilines.** Reactions were conducted on a 0.2 mmol scale under the standard conditions. Isolated yields are reported.

of stereoisomeric separation[50]. Further, the gram-scale synthesis of product **P51** (d.r. > 20:1) was successfully achieved by scaling up the reactants to 10 mmol, which still gave a desirable yield (Fig. 5c). Interestingly, representative compounds **P29** and **P51** underwent efficient debenzylation to afford N-unmasked products **P84** (d.r. > 20:1) and **P85** (d.r. > 20:1) in the presence of a Pd/HCOONH$_4$ system in methanol (Fig. 5d), which demonstrates the practicality of the developed chemistry in further preparation of fused heterocycles containing a useful −NH motif.

**Mechanistic investigations**. To gain mechanistic insights into the reaction, we conducted several control experiments (Fig. 6). First, the model reaction was interrupted after 6 hours to analyze the product system. Except for the formation of product **P1** in 23% yield, a dihydroquinoline **int-1** was isolated in 5% yield (Fig. 6a). Subjection of compound **int-1** (Fig. 6a and Supplementary Fig. 8) with aniline **B1** under the standard conditions resulted in product **P1** in high isolated yield (Fig. 6b), showing that **int-1** is a key reaction intermediate. However, removal of Ru-catalyst from the standard conditions failed to produce **P1** and the α-arylated product **P1'** (Fig. 6c), revealing that the reaction initiates with Ru-

catalyzed hydrogen transfer, instead of nucleophilic arylation of substrate **A1** with aniline **B1**. Further, the model reaction using deuterated methanol solvent yielded product **P1** without any D-incorporation (Fig. 6d). In sharp contrast, the same reaction by replacing paraformaldehyde with the fully deuterated one gave product **P1-d$_n$** with 35% and 28% D-ratios at the α and γ-sites and more than 99% D-ratio at the newly formed aminomethyl group (Fig. 6e and Supplementary Fig. 10). These two crucial experiments show that the formaldehyde serves as both the source of the reductant and C1-building block for the formation of the newly formed β-methylene group, and the initial reduction of **A1** to give either **int-1** or **int-2** is reversible (Fig. 1a). In parallel, we conducted the control experiments in terms of the generation of product **P51** (Supplementary Fig. 7 in SI). The results also support that dihydroquinoline **int-6** (Supplementary Fig. 7b) and β-methyl dihydroquinoline **int-9** (Supplementary Fig 9) are the reaction intermediates, and formaldehyde serves as the reductant source and C1-building block in the construction of the product (Supplementary Fig. 7d, e and Supplementary Fig. 11).

Based on the above findings, the plausible pathways toward the formation products **P1** and **P51** are depicted in Fig. 7. Initially, the

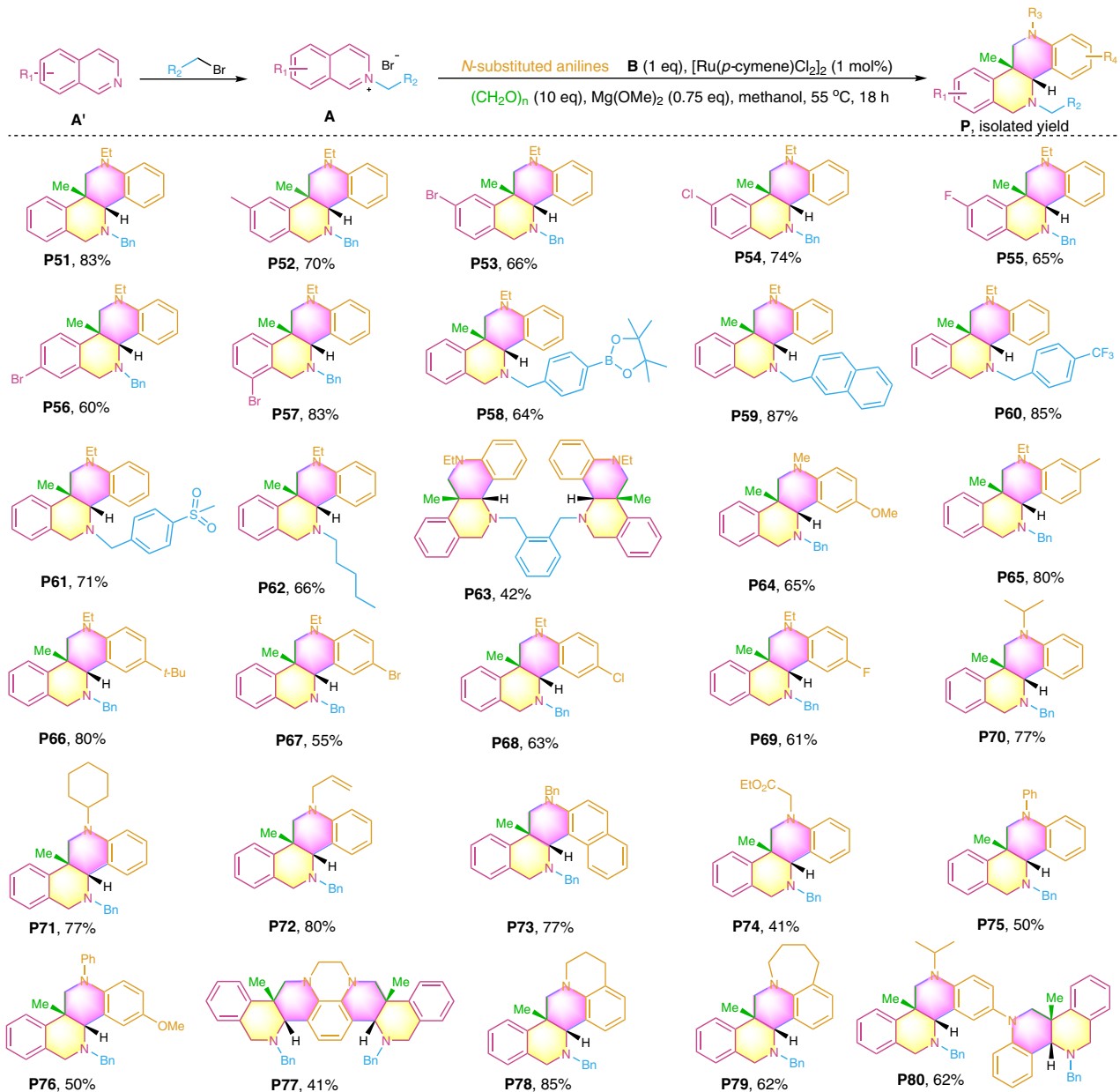

**Fig. 4 Diastereoselective access to fused N-heterocycles P51–P80 by employing various isoquinolinium salts.** Reactions were conducted on a 0.2 mmol scale under the standard conditions. Isolated yields are reported.

metal hydride species [Ru$^{II}$HX] is generated via Mg(OMe)$_2$ addition to formaldehyde (**E1**) followed by transmetallation (**E2**) with [Ru$^{II}$X$_2$] and $\beta$-hydride elimination and release of formate ester (detected by GC-MS analysis, Supplementary Fig. 12). Then, the hydride transfer from [Ru$^{II}$HX] to quinolinium salt **A1** forms dihydroquinoline **int-1** and its enamine tautomer **int-2** along with regeneration of the catalyst precursor [Ru$^{II}$X$_2$]. Meanwhile, the condensation between aniline **B1** and formaldehyde affords iminium **B1′**. Then, the $\beta$-nucleophilic addition of **int-2** to **B1′** gives the $\beta$-aminoalkyl iminium **int-3**. Further, the electron-rich benzene ring of **int-3** attacks the iminium motif from both the same (**int-3b**) and opposite (**int-3a**) sides of the H-atom at the $\beta$-site. In comparison, the form of **int-3a** (opposite side) is more favorable due to the less steric hindrance, thus affording product **P1** with *syn*-selectivity after deprotonation of the coupling adduct **int-4** (path a of Fig. 7b, namely electrophilic aryl C–H aminoalkylation). Alternatively, the [4 + 2] cycloaddition of **int-2** and **B1′** via *endo*

or *exo* π-π stacking also rationalizes the formation of **int-4** and product **P1** (path b of Fig. 7b, via **int-5** and **int −4**). Similarly, the generation of product **P51** from isoquinoline is shown in Fig. 6c. The hydride transfer from [Ru$^{II}$HX] to isoquinolinium salt **A32** initially forms enamine **int-6** (Supplementary Fig. 7a, b and Fig. 8). Then, the $\beta$-capture of formaldehyde by **int-6** followed by based-facilitated dehydration of **int-7** and hydride transfer to alkenyl iminium salt **int-8** forms $\beta$-methyl enamine **int-9** (Supplementary Fig. 7a and Fig. 9). Subsequently, the $\beta$-capture of **B1′** by **int-9** followed by intramolecular attack of the electron-rich phenyl ring to the iminium motif of **int-10** from the sterically less hindered back side of the methyl group, or the [4 + 2] cycloaddition of **int-9** and **B1′** via π-π stacking gives intermediate **int-11**. Finally, the deprotonation of **int-11** generates product **P51** with *syn*-diastereos-electivity (Fig. 7c).

To better reveal the selective formation of products **P1** and **P51**, computational study was performed using the density functional

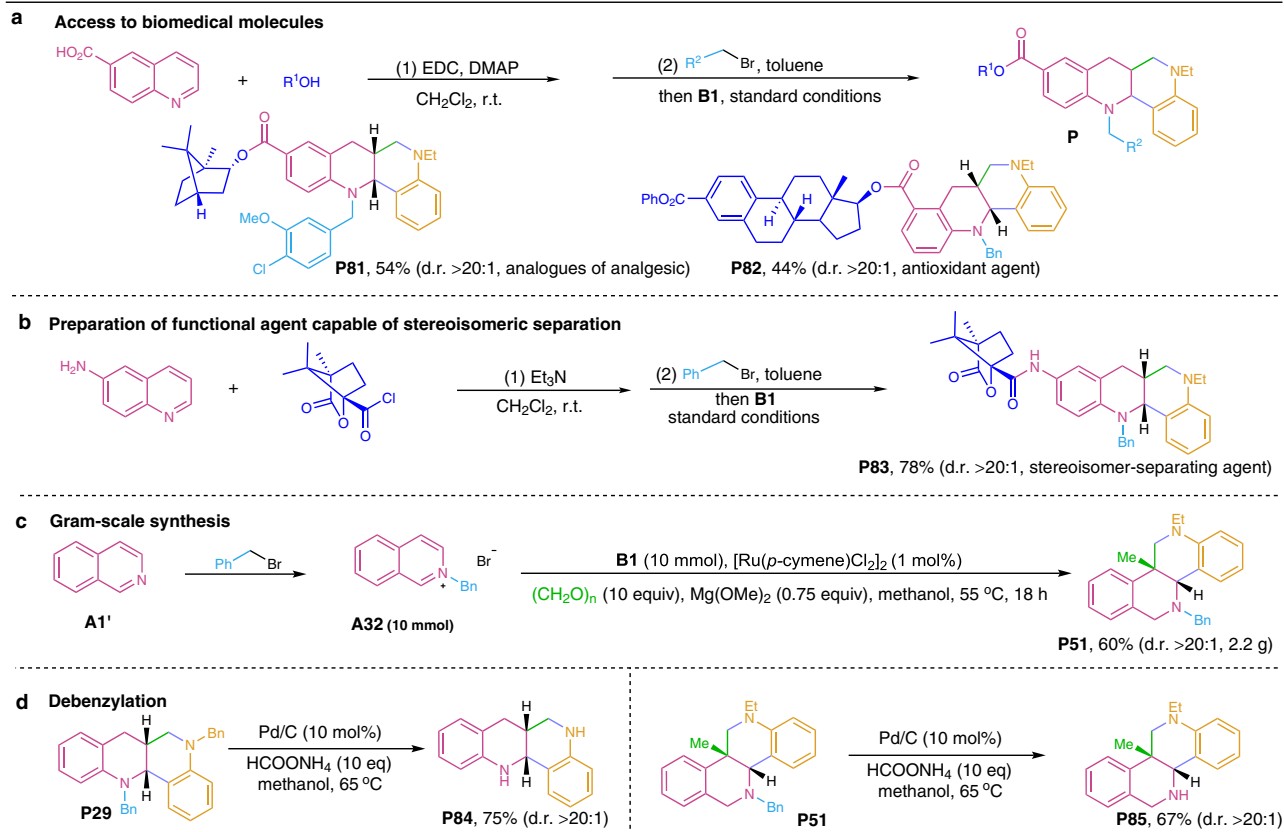

**Fig. 5 Synthetic applications of the developed chemistry. a, b** Structural modification of biomedical molecules. **c** Gram-scale synthesis. **d** Debenzylation of **P29** and **P51**.

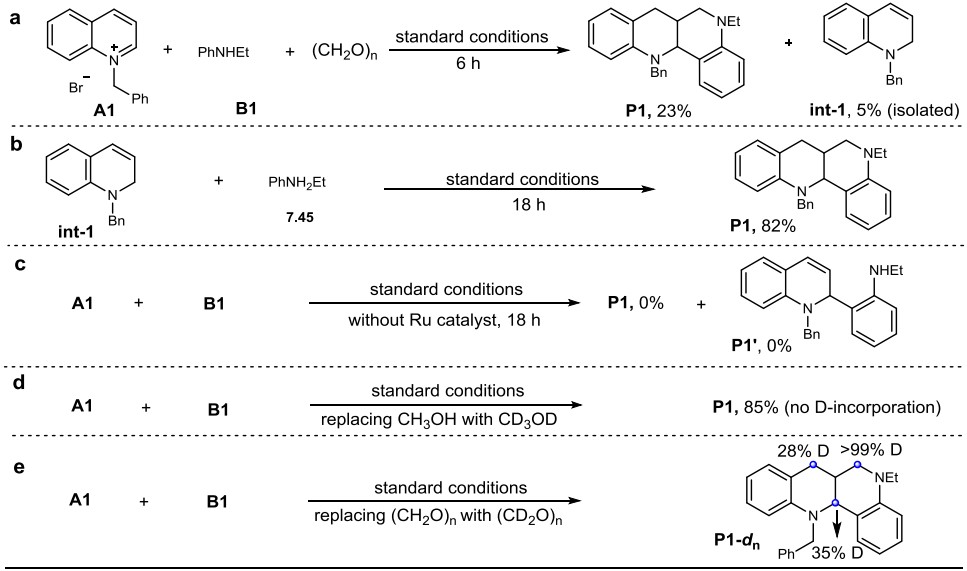

**Fig. 6 Control experiments for mechanistic studies. a** Intermediate Detection. **b** Intermediate verification. **c** Ruling out possible reaction step. **d, e** Deuterium-labeling experiments to verify hydrogen donor and C1 source.

theory and the relevant data was listed in the Supplementary Dataset file. First, the participation of Mg(OMe)₂, MeOK, and *t*-BuOK as the bases in the generation of [Ru$^{II}$HCl] was calculated. The energy for the transmetallation step with Mg(OMe)₂ (**int-22 → int-23**, $\Delta G = 10.1$ kcal mol⁻¹ in supplementary Fig. 13) is significantly higher than the other two bases (**int-26 → int-27**, $\Delta G = -3.6$ kcal mol⁻¹ for *t*-BuOK in supplementary Fig. 14; **int-**

**30 → int-23**, $\Delta G = -0.6$ kcal mol⁻¹ for MeOK in supplementary Fig. 15). The results reveal that the Mg²⁺ ion can better stabilize adduct **E1** (Fig. 7a) and make the dissociation of -MgOMe as well as the transmetallation process more difficult, thus resulting in a slow forming rate of [Ru$^{II}$HCl]. Correspondingly, a slow generation of enamine **int-2** via hydride transfer from [Ru$^{II}$HCl] to azaarenium salt **A1** is beneficial to the capture of **int-2** by **B1′**,

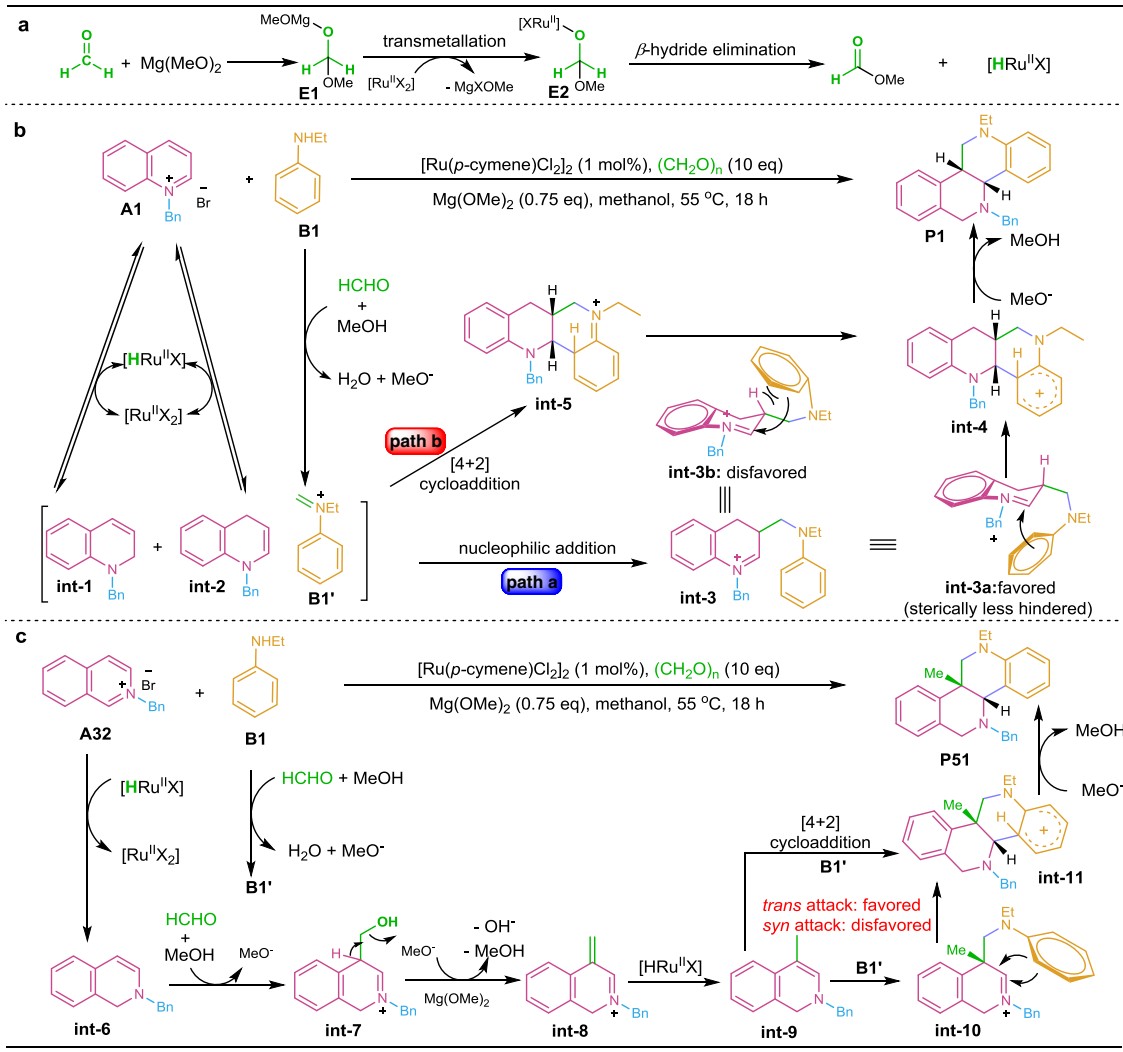

**Fig. 7 Plausible reaction mechanism. a** The production of metal hydride species. **b** Possible pathways for the formation product **P1**. **c** Possible pathway for the formation product **P51**.

and effectively suppresses the formation of undesired N-benzyl tetrahydroquinoline **A1″** (Table 1).

Next, the free energy profile for the conversion of **int-2** and **B1′** to **P1** is depicted in Fig. 8a and supplementary Fig. 16. The formation of **int-3a** and **int-3b** via β-nucleophilic addition of **int-2** to **B1′** has the energy barriers of 14.0 kcal mol$^{-1}$ (**TS4**) and 14.5 kcal mol$^{-1}$ (**TS4′**), respectively, representing endothermic processes ($\Delta G = 6.3$ kcal·mol$^{-1}$ and $\Delta G = 5.6$ kcal·mol$^{-1}$). The only difference between **int-3a** and **int-3b** is the dihedral of C2-C3-N2-C4 (**int-3a**, −76.0° vs **int-3b**, 106.9°), and the conversion of **int-3a** to **int-3b** is rather easy with only an energy barrier of 8.7 kcal·mol$^{-1}$. However, **int-3b** is not a favorable intermediate, as it has high stereoscopic hindrance of the N-ethyl group and pyridyl β-H as well as the long distance (~5.7 Å) between the pyridyl α-carbon (C1) and aniline *ortho*-carbon (C5). Therefore, **int-3a** becomes a favorable intermediate, and the attack of electron-rich aryl group to electron-deficient iminium motif generates **int-4** by intramolecular C1–C5 bond formation with an energy barrier of 15.5 kcal mol$^{-1}$ (**TS6**). Finally, the formation of product **P1**, via deprotonative aromatization of **int-4**, is a thermodynamically favorable process from the energetic point of view ($\Delta G = -54.1$ kcal mol$^{-1}$). In terms of the [4 + 2] cycloaddition of **B1′** and **int-2**, the manner of *endo* π-π stacking encountered commonly in the Diels-Alder reactions has a

significant energy barrier of 39.0 kcal mol$^{-1}$ (**TS7**). So, this pathway is disfavored. Meanwhile, the calculation results show that the manner of *exo* π-π stacking is also difficult to take place due to the higher steric hindrance and long interaction distance. Based on the computational studies, path a shown in Fig. 7b is believed to be a favorable way in generating product **P1**.

Further, the calculation results for the generation of product **P51** are depicted in Fig. 8b and supplementary Fig. 17. The formation of requisite intermediate **int-9** involves four main steps: (i) the β-addition of **int-6** to HCHO (**int-6 → int-6′**), (ii) proton transfer from the methanol (**int-6′ → int-7**), (iii) Mg(OMe)$_2$-induced proton abstraction and dissociation of OH$^-$ (**int-7 → int-7′ → int-8**), and (iv) hydride transfer (**int-8 → int-9**). Noteworthy, from step (i) to (iii), the formation of **int-8** clearly proceeds under the assistance of Mg$^{2+}$, and these steps can easily take place with a maximum energy barrier of 8.8 kcal·mol$^{-1}$ (**TS8**). Next, the formation of **int-9** by hydride transfer from [Ru$^{II}$HCl] complex to **int-8** proceeds with an energy barrier of 11.6 kcal mol$^{-1}$ (**TS11**), which is exothermic process ($\Delta G = -6.1$ kcal mol$^{-1}$). Once **int-9** is formed, the generation product **P51** proceeds with a similar way of Fig. 8a: β-addition of **int-9** to **B1′**, intramolecular cyclization via C1–C5 bond formation, and based-promoted deprotonation of **int-11** to yield product **P51**. The calculation results show that the processes from **int-9** to **P51** have a slightly higher barrier than that

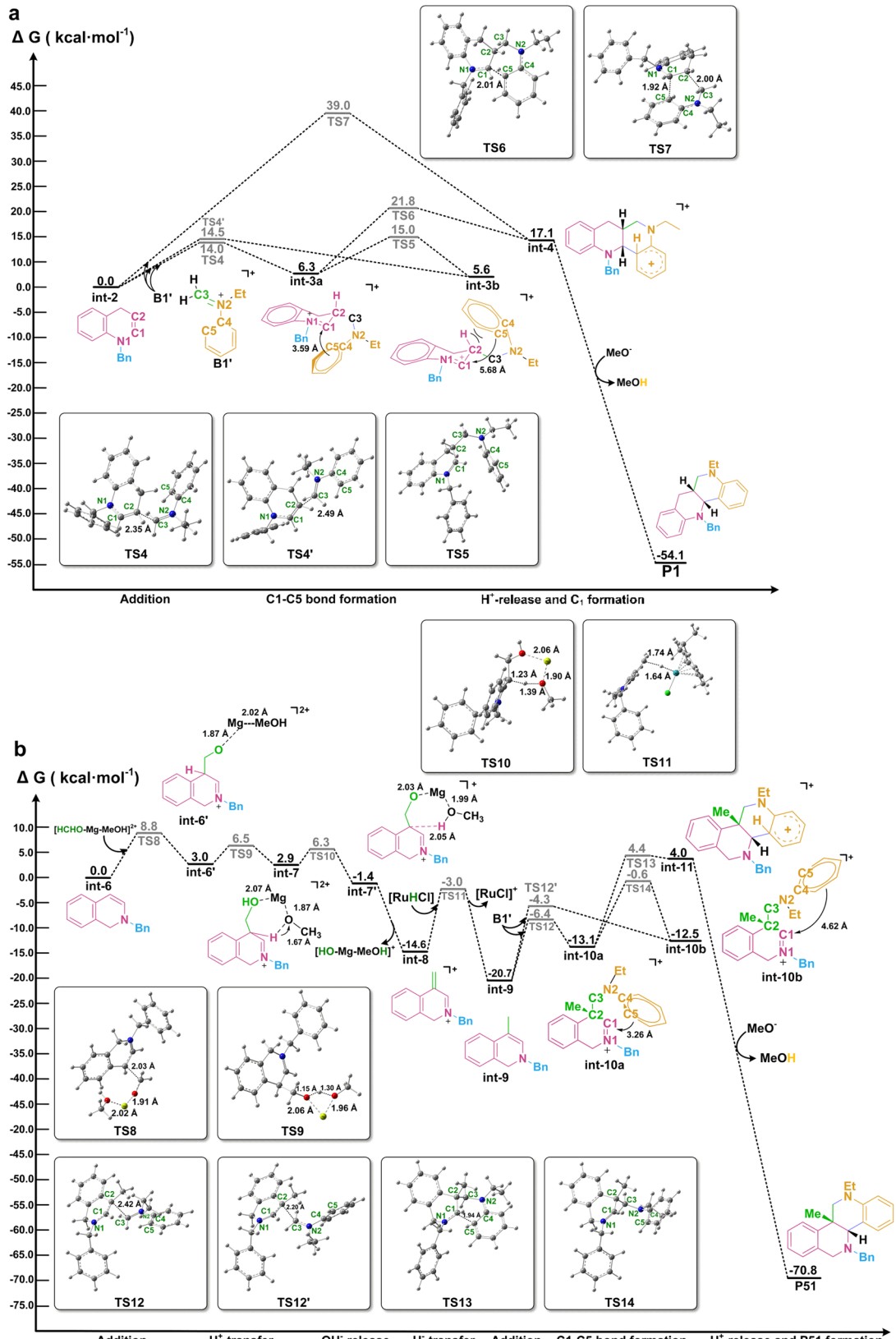

**Fig. 8 DFT studies (free energies in kcal mol⁻¹). a** Potential energy surfaces for the process from **int-2** to **P1 b** Potential energy surfaces for the process from **int-6** to **P51**.

of generating **P1** in Fig. 8a ($\Delta G^{\neq} = 17.5\,\text{kcal mol}^{-1}$ for **TS13** vs $\Delta G^{\neq} = 15.5\,\text{kcal mol}^{-1}$ for **TS6**).

In summary, by a strategy incorporating a tandem coupling sequence into the reduction of azaarenium salts, we have developed a intermolecular *syn*-diastereoselective annulation reaction by reductive ruthenium(II) catalysis. A variety of azaarenes were efficiently transformed in combination with a large variety of aniline derivatives into fused N-heterocycles by employing paraformaldehyde as both a crucial agent to generate active ruthenium(II)-hydride species and a C1-building block, proceeding with readily available feedstocks, excellent selectivity, mild conditions, and broad substrate and functional group compatibility. The present work has established a practical platform for the transformation of ubiquitously distributed but weakly reactive azaarenes into functional organic frameworks that are difficult accessible with the existing approaches, and further discovery of bioactive and drug-relevant molecules due to the promising potentials of the obtained compounds featuring the teterahydroquinolyl and hexahydro-1,6-naphthyridyl motifs. Mechanistic studies reveal that the products are formed via hydride transfer-initiated $\beta$-aminomethylation and $\alpha$-arylation of the azaarenium salts, and the use of Mg(OMe)$_2$ as a base plays a critical role in determining the reaction chemo-selectivity by lowering the hydrogen transfer rate. The work presented fills an important gap in the capabilities of utilizing azaarenes as the synthons to access fused N-heterocycles, and opens a door to further develop valuable reductive functionalization of inert unsaturated systems by taking profit of formaldehyde-endowed two functions.

## Methods

**Typical procedure I for the synthesis of product P1.** Under N$_2$ atmosphere, [Ru(*p*-cymene)Cl$_2$]$_2$ (1 mol %), 1-benzylquinolin-1-ium bromide **A1** (0.2 mmol), N-ethylaniline **B1** (0.2 mmol), Mg(OMe)$_2$ (0.75 eq, 12.9 mg), (CH$_2$O)$_n$ (10.0 eq, 60 mg) and methanol (1 mL) were introduced in a Schlenk tube, successively. Then the Schlenk tube was closed and the resulting mixture was stirred at 55 °C for 18 h. After cooling down to room temperature, the mixture was extracted with ethyl acetate, washed with 5% Na$_2$CO$_3$ solution, dried with anhydrous sodium sulfate, and then concentrated by removing the solvent under vacuum. Finally, the residue was purified by preparative TLC on silica to give 12-benzyl-5-ethyl-5,6,6a,7,12,12a-hexahydrodibenzo[*b*,*h*][1,6]naphthyridine **P1**.

See Supplementary methods for the structure and synthesis methods of the employed substrates, intermediates and utility. The analytical data and NMR spectra of all obtained compounds (**P1**–**P85**, **int-1** and **int-6**) are presented within Supplementary Information. See Supplementary Figs. 18–197 for NMR spectroscopic data of all compounds.

## Data availability

The X-ray crystallographic coordinates for structures reported in this article have been deposited at the Cambridge Crystallographic Data Centre (CCDC), under deposition number CCDC 2051637 (**P1**). The data can be obtained free of charge from The Cambridge Crystallographic Data Centre [http://www.ccdc.cam.ac.uk/data_request/cif]. The data supporting the findings of this study are available within the article and its Supplementary Information files. Any further relevant data are available from the authors on request.

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

## Acknowledgements

This paper is dedicated to Professor Matthias Beller on the occasion of his 60th birthday. We thank the National Natural Science Foundation of China (21971071, 22163007), Natural Science Foundation of Guangdong Province (2021A1515010155), the Fundamental Research Funds for the Central Universities (2020ZYGXZR075), and Guizhou Province Science Foundation ([2020]1Y050) for financial support.

## Author contributions

M.Z. conceived the idea, analyzed the data, directed the project, and wrote the manuscript. H.Z. and Y.W. carried out all the catalytic experiments. H.Z. drew the structures of all the obtained compounds and analyzed the single crystal structures. C.-G.C. performed the DFT calculations. Z.-D.T. and J.Y. synthesized the raw material. H.-F.J. and P.H.D. discussed the mechanistic aspects and revised the manuscript. All the authors have read the manuscript and agree with its content.

## Competing interests

The authors declare no competing interests.
