## [Peer Review File · Nature Communications]

REVIEWER COMMENTS

Reviewer #1 (Remarks to the Author):

In this manuscript Zhang and co-workers report an interesting tandem reduction and coupling of azarenes to form fused *syn-N*-heterocycles.

The mild reaction conditions use a low loadings of $[\text{Ru}(p\text{-cymene})\text{Cl}_2]_2$ as the catalyst, $\text{Mg}(\text{OMe})_2$ as a base, easily accessible aniline derivatives and paraformaldehyde as a cheap feedstock chemical. The paraformaldehyde serves a dual role being both the reductant precursor and the initial electrophile, as previously reported by Donohoe and co-workers (*Nat. Chem.* **2019**, *11*, 242–247, *Chem. Sci.*, **2021**, *12*, 742–746, etc.). The novelty of the proposed work is the β -aminomethylation followed by an α -arylation that introduces a tandem annulation step.

Overall the scope of the examples is impressive (80 examples in the scope) with mostly good to excellent yields across the board and good functional group tolerance. The relevant N-benzyl deprotections are highlighted and some pharmacologically interesting examples are presented in **Fig. 5**. The later two however lack a comment on diastereomeric ratios of products which would be expected given the reaction conditions.

Finally, extensive DFT calculations were performed to support the proposed mechanism following the formation of **int-2** and/or **int-6**, and to elucidate the role of $\text{Mg}(\text{OMe})_2$.

In summary, I am enthusiastic about publication of this excellent work in Nature Communications. I ask that the following points be addressed prior to publication:

1. The abstract should be slightly rewritten for greater clarity.
2. Have the authors considered the use of Rhodium catalysts (e.g.: $[\text{RhCp}^*\text{Cl}_2]_2$) to catalyse the transformation?
3. Have the authors considered the analogous reaction using pyridines as starting heterocycles?
4. I think that the proposed mechanism is inconsistent with the D-labeling experiments presented in the paper. The proposed equilibrium between **int-1** and **int-2** requires H/D transfer, however when the experiment is performed in CD_3OD no incorporation of deuterium is observed. Additionally, when $[\text{CD}_2\text{O}]_n$ is used the incorporation at the C-2 position is higher than in the C-4 position. The combination of these two results would suggest that the initial reduction of **A** to give either **int-1** or **int-2** is reversible and that there is no significant direct interconversion between **int-1** and **int-2**.

Therefore either additional data to support the mechanism presented throughout the manuscript needs to be provided or the mechanism needs to be altered to match the data currently available.

5. In **Fig.5** the reaction to give both **C₅₁** and **C₅₂** should result in a 1:1 mixture of diastereoisomers, however this fact is not addressed in the manuscript or the ESI.

Reviewer #2 (Remarks to the Author):

The authors describe their new synthetic method to construct fused syn-N-heterocycles from azaarenes and aniline derivatives. For this purpose, they employed a Ru(II) reductive catalytic system. DFT calculations will play key roles in deriving mechanistic insights into this kind of reaction. However, I have a strong concern about how DFT results are presented and discussed. I can imagine that the reader of this journal will not clearly understand what has been done in their computational study. Specific problems are summarized below. These issues have given me the impression that the work has not been done with sufficient care. Therefore, I do not recommend the publication of this manuscript in this journal.

(1) The first paragraph of the DFT calculations mentions the transmetallation barrier. For that, the reader is required to see Fig. S10-S12. However, there is no mention of "transmetallation" in the Supplementary Information (SI). Thus, the reader needs to do a lot of guesswork and find the corresponding step out of many other steps in Fig. S10-S12. This way of discussion is very confusing, and thus I suggest the authors revise the manuscript and SI accordingly. In addition, the author refers to Scheme 7a in the same paragraph. However, Scheme 7 cannot be found in the text.

(2) The reader will hardly understand what is presented in Fig. 8. The distance labels are too small, and the figure is blurred. Also, I notice that the energy scale in the DFT figure is not reasonable. For example, in Fig. 8b, the energy gap between Int-6 and TS8 (8.8 kcal/mol) is depicted as nearly the same as the gap between Int-6 and Int-9 (-20.7 kcal/mol). The bar's position for D1 (-70.8 kcal/mol) also seems to be too high. These problems with Fig. 8 will frustrate the reader, and at least, it was the case with myself. I didn't understand anything about Fig. 8 and the discussion in the corresponding paragraph. I cannot think this is the right way of preparing a manuscript.

(3) The Computational Details section in the SI is also very hard to follow. This is partly due to the language problem, and also because the figures were not prepared carefully.

Reviewer #3 (Remarks to the Author):

In this manuscript, Zhang and co-workers report a Ru-catalyzed hydride transfer-initiated annulation reaction between N-heteroarenium salts, anilines, and formaldehyde, enabling the construction of a wide range of fused N-heterocycles with excellent syn-diastereoselectivity, broad substrate scope, and good functional group tolerance under mild reaction conditions. The resulting products possess potential application for the discovery of functional molecules as their core structures are found in many bioactive and medicine-relevant compounds. Impressively, paraformaldehyde is utilized as both a crucial agent to generate reductive ruthenium(II)-hydride species and C1-building block. The use of Mg(OMe)₂ base plays a critical role in determining the reaction chemo-selectivity by lowering the hydrogen transfer rate. Moreover, this chemistry demonstrates a useful approach to transform less reactive pyridine-based azaarenes into functional molecular skeletons. Based on the control experiments and DFT calculations, possible reaction pathways are rationally proposed. This reviewer is very willing to recommend its publication in Nature Communications after addressing the following minor issues:

1. Although pyridine derivatives are less reactive, the authors should test the transformation of simple pyridine derivatives.

- 2. The authors should explain why the isoquinolinium salts react with formaldehyde firstly?**
- 3. Why methanol exhibits better performance than other alcohols or polar solvents?**
- 4. Authors should perform a thorough examination for the whole manuscript and reference as well as supporting information, as there are some typo errors, the writings to avoid typos.**

Reviewer #1:

*In this manuscript Zhang and co-workers report an interesting tandem reduction and coupling of azaarenes to form fused syn-N-heterocycles. The mild reaction conditions use a low loadings of [Ru(*p*-cymene)Cl₂]₂ as the catalyst, Mg(OMe)₂ as a base, easily accessible aniline derivatives and paraformaldehyde as a cheap feedstock chemical. The paraformaldehyde serves a dual role being both the reductant precursor and the initial electrophile, as previously reported by Donohoe and co-workers (Nat. Chem. 2019, 11, 242–247, Chem. Sci., 2021, 12, 742-746, etc.). The novelty of the proposed work is the β-aminomethylation followed by an α- arylation that introduces a tandem annulation step. Overall the scope of the examples is impressive (80 examples in the scope) with mostly good to excellent yields across the board and good functional group tolerance. The relevant N-benzyl deprotections are highlighted and some pharmacologically interesting examples are presented in Fig. 5. The later two however lack a comment on diastereomeric ratios of products which would be expected given the reaction conditions. Finally, extensive DFT calculations were performed to support the proposed mechanism following the formation of int-2 and/or int-6, and to elucidate the role of Mg(OMe)₂. In summary, I am enthusiastic about publication of this excellent work in Nature Communications. I ask that the following points be addressed prior to publication:*

Response: We sincerely thank you for your positive comments to our work.

1. The abstract should be slightly rewritten for greater clarity.

Response: Thank you for your constructive suggestion. We have rewritten the abstract.

*2. Have the authors considered the use of Rhodium catalysts (e.g.: [RhCp*Cl₂]₂) to catalyse the transformation?*

Response: Before submission of this manuscript, we have already tested the relevant catalysts such as Rh(OAc)₃ and [RhCp*Cl₂]₂, but they only gave low yields of product C₁ as compared to the use of [RuCl₂(*p*-cymene)]₂.

3. Have the authors considered the analogous reaction using pyridines as starting heterocycles?

Response: Thank you for your comments. However, before submission of this manuscript, we have performed such experiment by using 4-phenyl pyridine as starting heterocycles, but it failed to give the desired product, and only β-aminoalkyl product (Fig. 3, C₅₀) was obtained. The result indicates that arene-fused pyridyl substrates are essential for the success of the annulation reaction.

4. I think that the proposed mechanism is inconsistent with the D-labeling experiments presented in the paper. The proposed equilibrium between int-1 and int-2 requires H/D transfer, however when the experiment is performed in CD₃OD no incorporation of deuterium is observed. Additionally, when [CD₂O]_n is used the incorporation at the C-2 position is higher than in the C-4 position. The combination of these two results would suggest that the initial reduction of A to give either int-1 or

int-2 is reversible and that there is no significant direct interconversion between int-1 and int-2.

Therefore either additional data to support the mechanism presented throughout the manuscript needs to be provided or the mechanism needs to be altered to match the data currently available.

Response: Many thanks for the constructive comments. Your interpretation is quite reasonable. So, we have altered the mechanism in Fig. 7, and the relevant expression have also been revised in the paragraph above Fig. 6 in the manuscript.

5. In Fig.5 the reaction to give both C₅₁ and C₅₂ should result in a 1:1 mixture of diastereoisomers, however this fact is not addressed in the manuscript or the ESI.

Response: Many thanks for pointing out the issue. We have supplemented the diastereoselective value for products C₅₁-C₅₄ and D₃₁ in the manuscript (see Fig. 5 and the related description).

Reviewer #2:

The authors describe their new synthetic method to construct fused syn-N-heterocycles from azaarenes and aniline derivatives. For this purpose, they employed a Ru(II) reductive catalytic system. DFT calculations will play key roles in deriving mechanistic insights into this kind of reaction. However, I have a strong concern about how DFT results are presented and discussed. I can imagine that the reader of this journal will not clearly understand what has been done in their computational study. Specific problems are summarized below. These issues have given me the impression that the work has not been done with sufficient care. Therefore, I do not recommend the publication of this manuscript in this journal.

Response: Thank you very much for your comments and pointing out the issues. We have performed a careful revision for the whole manuscript and supporting information according to your comments, and we believe the DFT results are now clearly presented.

1. The first paragraph of the DFT calculations mentions the transmetallation barrier. For that, the reader is required to see Fig. S10-S12. However, there is no mention of “transmetallation” in the Supplementary Information (SI). Thus, the reader needs to do a lot of guesswork and find the corresponding step out of many other steps in Fig. S10-S12. This way of discussion is very confusing, and thus I suggest the authors revise the manuscript and SI accordingly. In addition, the author refers to Scheme 7a in the same paragraph. However, Scheme 7 cannot be found in the text.

Response: We sincerely thank you for pointing out the issues. (a) “transmetallation” has been supplemented in Fig. S10-S12 in the supporting information, and these figures are now clear to understand for the readers. Moreover, we have revised all the figures in both the manuscript and supporting information. (b) The energy for the transmetallation step with Mg(OMe)₂ (**int-22** → **int-23**, $\Delta G = 10.1 \text{ kcal}\cdot\text{mol}^{-1}$, Fig. S10) is significantly higher than the other two bases (**int-26** → **int-27**, $\Delta G = -3.6 \text{ kcal}\cdot\text{mol}^{-1}$ for *t*-BuOK, Fig. S11 and **int-30** → **int-23**, $\Delta G = -0.6 \text{ kcal}\cdot\text{mol}^{-1}$ for MeOK, Fig. S12) in the potential energy surfaces.

We are sorry for the carelessness, Scheme 7a has been corrected as Fig. 7a.

2. The reader will hardly understand what is presented in Fig. 8. The distance labels are too small, and the figure is blurred. Also, I notice that the energy scale in the DFT figure is not reasonable. For example, in Fig. 8b, the energy gap between Int-6 and TS8 (8.8 kcal/mol) is depicted as nearly the same as the gap between Int-6 and Int-9 (-20.7 kcal/mol). The bar’s position for DI (-70.8 kcal/mol) also seems to be too high. These problems with Fig. 8 will frustrate the reader, and at least, it was the case with myself. I didn’t understand anything about Fig. 8 and the discussion in the corresponding paragraph. I cannot think this is the right way of preparing a manuscript.

Response: The Fig. 8 has been redrawn. Thank you very much for pointing out the issues, we have rewritten the discussion.

3. *The Computational Details section in the SI is also very hard to follow. This is partly due to the language problem, and also because the figures were not prepared carefully.*

Response: Many thanks for pointing out the issues. We have rewritten the figures and carefully revised the computational details in the SI according to your suggestions.

Reviewer #3:

In this manuscript, Zhang and co-workers report a Ru-catalyzed hydride transfer-initiated annulation reaction between N-heteroarenium salts, anilines, and formaldehyde, enabling the construction of a wide range of fused N-heterocycles with excellent syn-diastereoselectivity, broad substrate scope, and good functional group tolerance under mild reaction conditions. The resulting products possess potential application for the discovery of functional molecules as their core structures are found in many bioactive and medicine-relevant compounds. Impressively, paraformaldehyde is utilized as both a crucial agent to generate reductive ruthenium(II)-hydride species and C1-building block. The use of Mg(OMe)₂ base plays a critical role in determining the reaction chemo-selectivity by lowering the hydrogen transfer rate. Moreover, this chemistry demonstrates a useful approach to transform less reactive pyridine-based azaarenes into functional molecular skeletons. Based on the control experiments and DFT calculations, possible reaction pathways are rationally proposed. This reviewer is very willing to recommend its publication in Nature Communications after addressing the following minor issues:

Response: We sincerely thank you for your positive comments to our work.

1. *Although pyridine derivatives are less reactive, the authors should test the transformation of simple pyridine derivatives.*

Response: Many thanks for your constructive comments. However, before submission of this manuscript, we have performed such experiment by using 4-phenyl pyridine as starting heterocycles, but it failed to give the desired product, and only β -aminoalkyl product (Fig. 3, **C₅₀**) was obtained. The result indicates that arene-fused pyridyl substrates are essential for the success of the annulation reaction.

2. *The authors should explain why the isoquinoliunium salts react with formaldehyde firstly?*

Response: Many thanks for the nice question. First, by interruption of the model reaction after 3 h under the optimal conditions, the β -methyl dihydroquinoline (**int-9**) can be detected by GC-MS (see Scheme S3a). After completion of the reaction, **int-9** is fully consumed up, indicating that **int-9** is a key reaction intermediate. Moreover, the DFT calculations explain why the isoquinoliunium salts preferentially react with formaldehyde (see Fig.8b and the related description in the manuscript; Fig. S14 and the related description in supporting information).

3. Why methanol exhibits better performance than other alcohols or polar solvents?

Response: As compared to other alcohols or polar solvents, we observed that the use of methanol is best choice to dissolve $\text{Mg}(\text{OMe})_2$, whereas $\text{Mg}(\text{OMe})_2$ base plays a critical role in determining the reaction chemo-selectivity by lowering the hydrogen transfer rate.

4. Authors should perform a thorough examination for the whole manuscript and reference as well as supporting information, as there are some typo errors, the writings to avoid typos.

Response: Many thanks for your nice suggestion. We have performed a thorough examination for the whole manuscript and reference as well as supporting information and corrected the errors.

Reviewer #1 (Remarks to the Author):

I am happy that my comments have been addressed in full, and so I recommend publication without changes.

Reviewer #2 (Remarks to the Author):

The manuscript has been revised reasonably well, although many of the labels in Fig. 8 are too small to read.

Reviewer #3 (Remarks to the Author):

The authors have responded to the comments and made the corresponding revisions, therefore, this reviewer recommends its publication as it stands.

Point-to-point response letter

(Manuscript ID: NCOMMS-21-01378, Zhang et al.)

Reviewer #1 (Remarks to the Author):

I am happy that my comments have been addressed in full, and so I recommend publication without changes.

Response: Thank you for your support in publication of our work.

Reviewer #2 (Remarks to the Author):

The manuscript has been revised reasonably well, although many of the labels in Fig. 8 are too small to read.

Response: Thank you for your support in publication of our work. We have enlarged the labels in Fig. 8 according to your suggestion.

Reviewer #3 (Remarks to the Author):

The authors have responded to the comments and made the corresponding revisions, therefore, this reviewer recommends its publication as it stands.

Response: Thank you for your support in publication of our work.